# Neutrophils actively swell to potentiate rapid migration

Tamas L Nagy[1,2]*[†], Evelyn Strickland[1,2], Orion D Weiner[1,2]*

[1]Department of Biochemistry and Biophysics, University of California, San Francisco, San Francisco, United States; [2]Cardiovascular Research Institute, University of California, San Francisco, San Francisco, United States

**Abstract** While the involvement of actin polymerization in cell migration is well-established, much less is known about the role of transmembrane water flow in cell motility. Here, we investigate the role of water influx in a prototypical migrating cell, the neutrophil, which undergoes rapid, directed movement to sites of injury, and infection. Chemoattractant exposure both increases cell volume and potentiates migration, but the causal link between these processes are not known. We combine single-cell volume measurements and a genome-wide CRISPR screen to identify the regulators of chemoattractant-induced neutrophil swelling, including NHE1, AE2, PI3K-gamma, and CA2. Through NHE1 inhibition in primary human neutrophils, we show that cell swelling is both necessary and sufficient for the potentiation of migration following chemoattractant stimulation. Our data demonstrate that chemoattractant-driven cell swelling complements cytoskeletal rearrangements to enhance migration speed.

## eLife assessment

This **fundamental** study significantly advances our understanding of the role of water influx and swelling on neutrophil migration in response to chemoattractant. The evidence supporting the conclusions, based on a genome-wide CRISPR screen and high quality cellular observations, is **compelling**. This paper will be of interest to cell biologists and biophysicists working on cell migration.

## Introduction

Cells extend their membranes during morphogenesis and movement. Membrane extension in plant and fungal cells are thought to require coordination between water influx and cell wall remodeling (*Boyer and Silk, 2004*; *Lew, 2011*). In animal cells that lack cell walls, most research has focused on the role of cytoskeletal rearrangements during membrane extension and migration (*Svitkina, 2018*), though in some contexts water influx also appears to suffice for cell movement (*Stroka et al., 2014*; *Zhang et al., 2022*). It is likely that cytoskeletal rearrangements and water influx collaborate in a wider range of cell morphological and migratory contexts than are currently appreciated.

In contrast to the extensive work characterizing the actin regulators that underlie cell motility, we have a relatively poor understanding of the molecular regulators of water influx during cell migration. We address this question in neutrophils, which are innate immune cells that polarize their actin assembly and actomyosin contractility to move to sites of injury and infection (*Weiner et al., 1999*; *Sengupta et al., 2006*; *Lämmermann et al., 2013*). Chemoattractant stimulation also initiates water influx and cell swelling (*Grinstein et al., 1986*), but the molecular basis of this swelling and its relevance to motility are not known.

**\*For correspondence:**
iam@tamasnagy.com (TLN);
orion.weiner@ucsf.edu (ODW)

**Present address:** [†]Department of Neurology, University of California, Los Angeles, Los Angeles, United States

**Competing interest:** The authors declare that no competing interests exist.

## Results

### Chemoattractant stimulation elicits competing volume responses in primary human neutrophils

Neutrophils are a powerful system to study the biophysical demands of cell motility, as they acutely initiate rapid migration following stimulation with a chemoattractant. Normally, they exist in a quiescent non-motile state (*Metzemaekers et al., 2020*). Upon exposure to chemoattractants, neutrophils respond with significant morphological changes (*Sengupta et al., 2006*; *Denk et al., 2017*), water influx (*Grinstein et al., 1986*; *Pember et al., 1983*), and a dramatic increase in motility (*Martin et al., 2015*).

To probe the role of transmembrane water flow in chemoattractant-stimulated morphogenesis and movement, we adapted Fluorescence eXclusion Microscopy (FxM) (*Cadart et al., 2017*), a single-cell volume measurement technique, to primary human neutrophils (*Figure 1A*; *Figure 1—figure supplement 1A–E*). This assay enables us to accurately measure absolute cell volume in single primary human neutrophils during activation by chemoattractant (*Figure 1B–C*). We used a UV-sensitive caged chemoattractant (*Collins et al., 2015*) to acutely stimulate the cells in situ with a short (10 s) pulse of light enabling us to capture both the volume and motility response of cells before and after activation. Following chemoattractant-uncaging, the cells spread and transformed into a motile, amoeboid state with high persistence (*Figure 1C*). By simultaneously measuring the single-cell volumes, we observed a biphasic volume response (*Figure 1D*; *Video 1*). Immediately following chemoattractant exposure, the cells lose 5–8% of their cell volume (*Figure 1D*, inset), consistent with spreading-induced volume losses previously observed in several other cell types (*Venkova et al., 2022*). Following this spreading-mediated initial volume loss, the cells swelled significantly, reaching a median volume 15% larger than their resting volumes after 20 minutes, agreeing with previous reports (*Grinstein et al., 1986*; *Pember et al., 1983*). The chemoattractant-induced spreading can be seen by the increased cell footprint area post-stimulation (*Figure 1E*). This raises the question of whether the swelling is downstream of spreading and motility. To address this point, we used Latrunculin B to depolymerize the actin cytoskeleton and found that while chemoattractant-induced swelling is unaffected, motility was completely blocked (*Figure 1—figure supplement 2A–E*; *Figure 1—video 1*). These data suggest that while chemoattractant simultaneously drives the cell volume increase and the motility changes, these processes are separable from one another.

While the volume of the median cell increases significantly following chemoattractant stimulation, this masks more complex behavior at the single-cell level. Single-cell analyses reveal that even as the baseline volume has increased post-activation, individual cells exhibit large fluctuations in volume on the single-minute time scale as they move (*Figure 1F*; *Video 2*). These fluctuations appear correlated with the neutrophil motility cycle and require an intact actin polymerized cytoskeleton (*Figure 1—figure supplement 2F, G*). The increase in the volume set point (from 2 mins to 20–30 min) is closely correlated with the increases in cell velocities over the same time frame (*Figure 1G*). To investigate whether there is a causal link between the chemoattractant-induced cell swelling and migration potentiation, we next sought to identify the molecular regulators of neutrophil swelling.

### Genome-wide screen identifies regulators of chemoattractant-induced cell swelling

As an unbiased approach for identifying the regulators of chemoattractant-induced cell swelling, we turned to pooled genome-wide CRISPR/Cas9 screening (*Shalem et al., 2014*). Our approach relies on creating a population of cells with single gene knockouts and then enriching for the cells that fail to swell following stimulation. The quality of this enrichment is the most critical step for success of these screens (*Nagy and Kampmann, 2017*). A key challenge in adapting pooled CRISPR screening to this context was the lack of highly scalable approaches for accurately separating the cells based on their volumes directly. Although volume is difficult to use as a separation approach, cells can be easily separated by buoyant density (mass over volume), which is related to volume over short timescales. Because neutrophil swelling in suspension results from the uptake of water (*Grinstein et al., 1986*), stimulated neutrophils exhibit a corresponding decrease in buoyant density (*Pember et al., 1983*). Buoyant density has been successfully used in other genetic screens, including the identification of secretion-defective mutants in yeast (*Novick et al., 1980*). Finally, buoyant density is a particularly

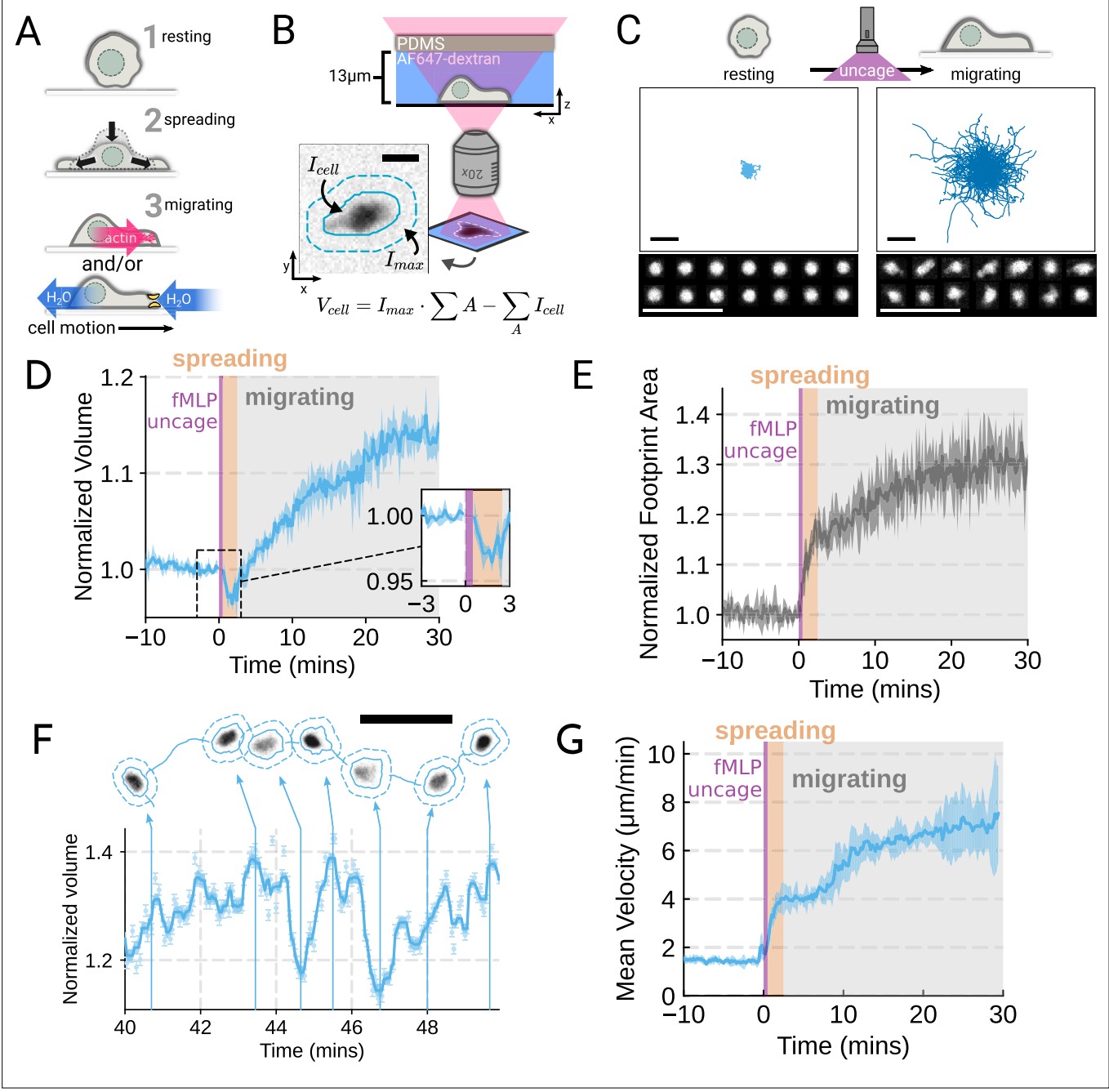

**Figure 1.** Chemoattractant stimulation elicits competing volume responses in primary human neutrophils (**A**) Schematic detailing the neutrophil activation process. (**B**) Schematic detailing the Fluorescence Exclusion Microscopy (FxM) approach for measuring single-cell volumes, which relies on cells displacing an extracellular dye in a shallow microfluidic chamber. The inset shows an example cell with the cell footprint and local background indicated by the solid or dashed teal lines, respectively. The scale bar is 10 µm. (**C**) Primary human neutrophil tracks over 15 min time windows before (left) and after (right) the uncaging of the fMLP chemoattractant. Randomly selected example cells in the bottom panel show neutrophil shape before and after activation. All scale bars are 50 µm. (**D**) Normalized volumes of primary human neutrophils in response to chemoattractant stimulation (Volunteer n = 4, Cells = 440 total). Inset details the volume loss due to spreading immediately after uncaging. Cells initially lose volume during the spreading phase following the chemoattractant stimulation and then significantly increase in volume. The line plotted is the average of the median cell response for each volunteer, and the shaded region is the 95% CI of the mean. See *Video 1* for an animated version. (**E**) The normalized footprint area of primary human neutrophils responding to chemoattractant. The line is the average across biological replicates of median cell footprint area at each timepoint. The footprint area shows a monotonic increase in response to activation with cell spreading prior to initiation of movement. (**F**) Single representative cell trace imaged with high time resolution to highlight the cell motility-related volume fluctuations. Top section depicts the cell track with the FxM images overlaid at key time points that are linked with cyan arrows to the corresponding volumes in the bottom plot. Bottom section is a

*Figure 1 continued on next page*

*Figure 1 continued*

scatter plot of the raw volume values, with the thick cyan line depicting the rolling median volume. See *Video 2* for an animated version. Scale bar is 50 µm. (**G**) Mean of the per-replicate median cell velocities computed at each time point. The shaded area is the standard deviation at each time point. Cell migration begins to increase in the early spreading phase following chemoattractant stimulation and then continues to increase over the next 20 min following stimulation.

The online version of this article includes the following video and figure supplement(s) for figure 1:

**Figure supplement 1.** Details and validation of the Fluorescence eXclusion Microscopy pipeline.

**Figure supplement 2.** Chemoattractant-induced swelling, but not motility, is independent of actin polymerization.

**Figure 1—video 1.** Inhibiting actin polymerization blocks neutrophil chemokinesis.

https://elifesciences.org/articles/90551/figures#fig1video1

homogenous parameter at the population-level, with 100-fold less variation than either mass or volume across multiple different cell types (*Grover et al., 2011*).

To verify the chemoattractant-induced shifts in neutrophil buoyant density in our own hands, we deposited linear Percoll gradients (*Figure 2—figure supplement 1A–B*) in centrifuge tubes and carefully layered nutridoma-differentiated HL-60s (dHL-60s) onto the gradients in the absence or presence of the chemoattractant fMLP (*Figure 2A*). Stimulating dHL-60s with fMLP and using an optimized centrifugation protocol (*Figure 2—figure supplement 1C*) led to a robust, long-term decrease in buoyant density across millions of cells with a shift in population position clearly visible by eye (*Figure 2B*). The buoyant density change corresponded to a 15% increase in cell volume (*Figure 2C*). This effect depends on chemoattractant-based cell stimulation, as knockout of the fMLP receptor FPR1 completely inhibits fMLP-induced swelling (*Figure 2C*, right).

To screen for the chemoattractant-induced regulators of cell volume, we transduced HL-60 cells with a commercial genome-wide CRISPR knockout library, differentiated them, and spun the cell population into Percoll density gradients with or without fMLP stimulation. We then fractionated the tubes and partitioned the samples into three different groups: low, medium, and high buoyant density (*Figure 2—figure supplement 1D*). We used next-generation sequencing to determine which CRISPR guides were over-represented in the high-density bin, i.e., which guides prevented swelling, leading to the cells remaining dense following stimulation with fMLP (*Figure 2D*).

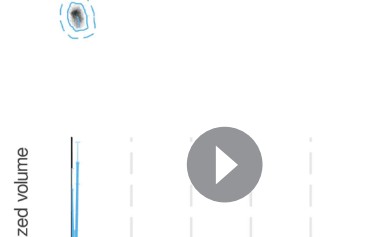

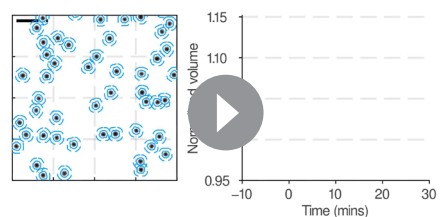

**Video 1.** Primary human neutrophils show a biphasic volume response to chemoattractant stimulation. Left panel shows the individual cells (cyan solid lines) with their individual localities (dashed lines). The actual Fluorescence eXclusion Microscopy (FxM) signal is visible within the cell footprints. The video is 40 min long with chemoattractant uncaging after the first 10 min. The scale bar is 50 µm. The right panel shows the corresponding median volume of the population (cyan) with the 95% confidence interval (light cyan). The volume is relative to the average over the 2 min window prior to uncaging. Data available via Dryad; see https://doi.org/10.7272/Q6NS0S5N.

https://elifesciences.org/articles/90551/figures#video1

**Video 2.** Single-cell volume tracking reveals motility-associated volume changes in migrating primary human neutrophils. Top panel shows a representative cell migrating starting at 40 min following chemoattractant stimulation. The solid cyan line is the cell track. The cell's footprint is denoted with a solid cyan line while its local background is encapsulated by the dashed cyan line. The scale bar is 50 µm. The bottom panel shows the concurrent volume changes with the raw Fluorescence eXclusion Microscopy (FxM) measurements displayed in light cyan and the rolling median is shown in cyan. The volume is normalized to the median cell volume in the 2 min window prior to stimulation. Data available via Dryad; see https://doi.org/10.7272/Q6NS0S5N.

https://elifesciences.org/articles/90551/figures#video2

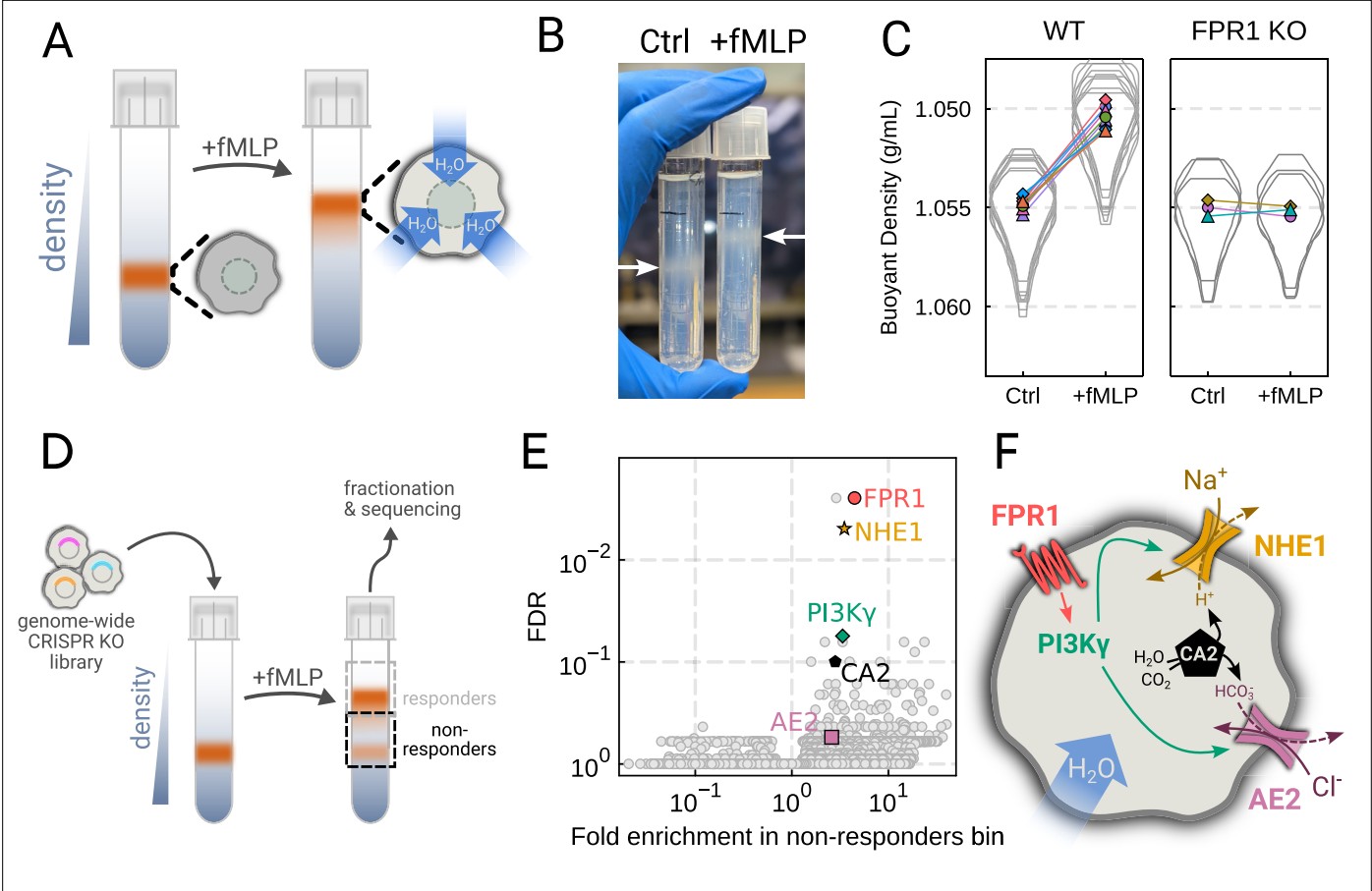

**Figure 2.** Genome-wide screen identifies regulators of chemoattractant-induced cell swelling. (**A**) Schematic detailing the buoyant density assay. The addition of fMLP causes cells to swell and decrease their density. As a result, the stimulated cells float higher in the Percoll density gradient. (**B**) Representative image of cell density shift following chemoattractant stimulation. Millions of cells appear as white fuzzy bands (indicated with the arrows). Cells in the right tube are stimulated with 20 nM chemoattractant (fMLP), causing them to swell and float higher in the gradient. (**C**) Violin plots quantifying the relative cell numbers as a function of density. Individual lines link replicate pairs. Wild-type (WT) cells shift from 1.055 g/mL to 1.050 g/mL upon stimulation, while FPR1 KO cells do not shift following fMLP stimulation. (**D**) Schematic detailing the buoyant density-based genome-wide CRISPR knockout screen for identifying cells that are deficient at chemoattractant-induced cell swelling. (**E**) Volcano plot of the results of the chemoattractant-induced cell swelling screen. Genes that showed large inhibition of cell swelling and consistent behavior across their targeting guides appear in the upper right. The genes selected for further analysis are highlighted for a more complete list see *Supplementary file 1A*. (**F**) Schematic outlining a potential pathway from chemoattractant stimulation to cell swelling.

The online version of this article includes the following figure supplement(s) for figure 2:

**Figure supplement 1.** Validation of buoyant density assay and its use in CRISPR KO genome-wide screen for swelling regulators.

To verify CRISPR knockout efficacy, we confirmed the systematic depletion of essential genes from the population (*Figure 2—figure supplement 1E*). Computing median log2-fold enrichment of the guides targeting each gene in the dense bin and plotting this value against the false discovery rate revealed the regulators of chemoattractant-induced cell swelling (*Figure 2E*). The top right corner is occupied by genes that are over-represented in the dense, i.e., non-swelling, bin. The top hit was FPR1, the high affinity GPCR that specifically binds to fMLP to initiate the chemoattractant signaling cascade, confirming the effectiveness of the screen.

Our screen revealed a potential transduction cascade from the chemoattractant receptor to the final effectors of cell swelling, including the sodium-proton antiporter NHE1 (SLC9A1), the chloride-bicarbonate exchanger 2 (AE2, i.e. SLC4A2), the gamma subunit of phosphoinositide 3-kinase (PI3Kγ), and carbonic anhydrase II (CA2) (*Figure 2F*). These hits suggest that the cell swelling cascade begins with fMLP binding to the chemoattractant receptor FPR1, which activates PI3Kγ, which in turn activates NHE1 and AE2, the canonical regulatory volume increase complex (*Hoffmann et al., 2009*).

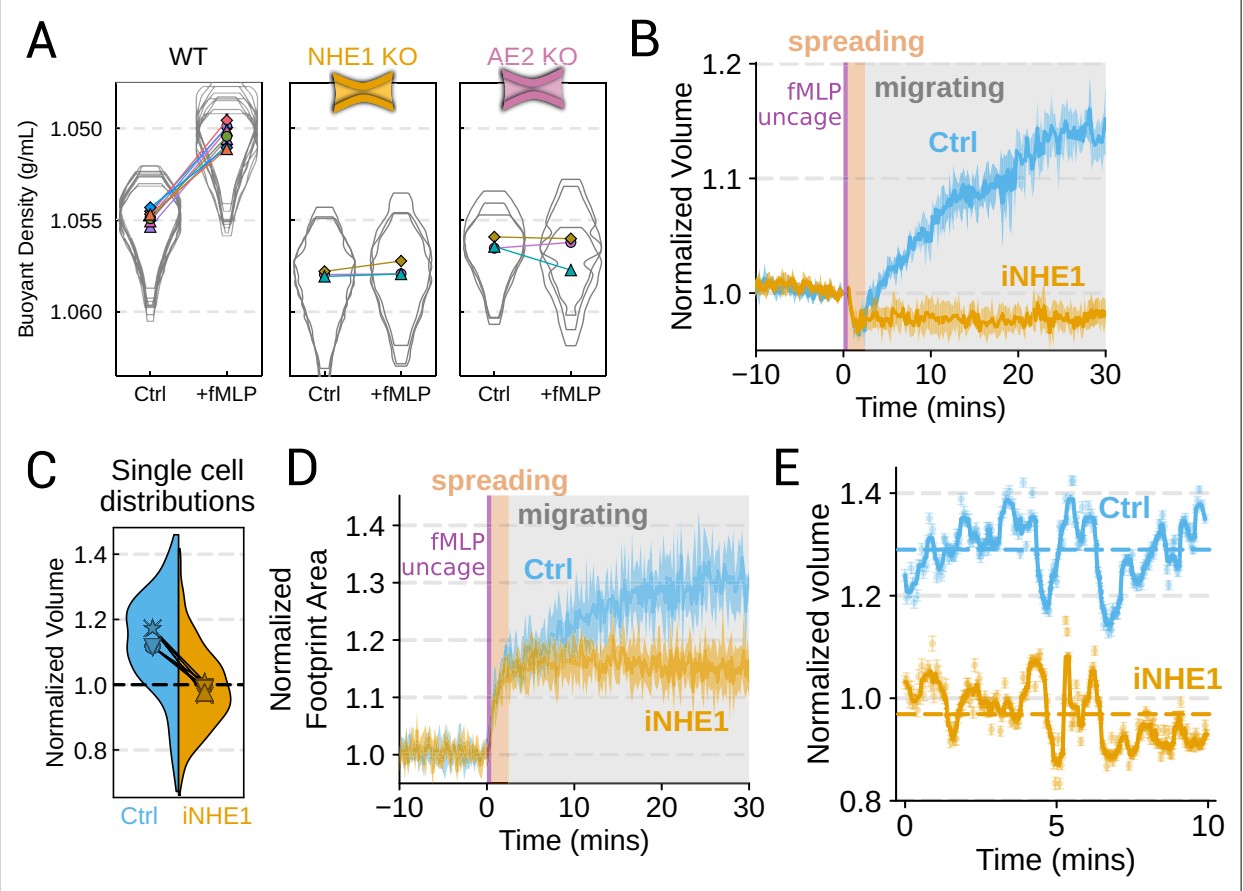

**Figure 3.** Mechanistically separating chemoattractant versus motility-based volume changes (**A**) Knockout of NHE1 or AE2 completely inhibits the chemoattractant-induced swelling in neutrophil-like differentiated HL-60 cells. (**B**) NHE1 inhibition in human primary neutrophils blocks fMLP-induced swelling but does not inhibit the spreading-induced volume loss. An animated version is available as *Video 4*. (Ctrl: Volunteer N = 4, iNHE1: Volunteer n = 6). (**C**) The distributions of single-cell volumes 30 min post-chemoattractant stimulation demonstrates that the NHE1-inhibited neutrophils remain close to the pre-stimulation volumes, i.e., 1.0 on the ordinate. (**D**) NHE1-inhibited neutrophils have similar increases in their footprint area when they spread and begin moving following fMLP stimulation. (**E**) High temporal resolution imaging of the motility-induced volume fluctuations starting at 30 min post-stimulation demonstrate that both control and NHE1-inhibited neutrophils show similar short-term volume fluctuations around significantly different baselines (dashed lines). For an animated version, see *Videos 2 and 3*.

The online version of this article includes the following figure supplement(s) for figure 3:

**Figure supplement 1.** Additional validation of swelling screen hits.

NHE1 and AE2 would, in this model, eject cytoplasmic protons and bicarbonate ions in exchange for extracellular sodium and chloride, respectively. CA2 catalyzes the production of protons and bicarbonate from $CO_2$ and water and has previously been reported to bind the tail of NHE1, enhancing its activity (*Li et al., 2002*; *Li et al., 2006*). Thus, fMLP binding would lead to a net influx of sodium and chloride into the cell, mediating the influx of water and resulting in cell swelling.

## Mechanistically separating chemoattractant versus motility-based volume changes

We next sought to individually validate our hits for chemoattractant-induced swelling. We created and verified single gene knockouts of the four components–NHE1, AE2, PI3Kγ, and CA2–using CRISPR/Cas9 in HL-60 cells. Using our buoyant density assay, we found that loss of either NHE1 or AE2 completely ablated the fMLP-induced volume increase in dHL-60s (*Figure 3A*). Our data indicate that both ion channels are needed for chemoattractant-induced swelling. Knockouts of PI3Kγ and CA2 partially inhibited chemoattractant-induced swelling (*Figure 3—figure supplement 1A*).

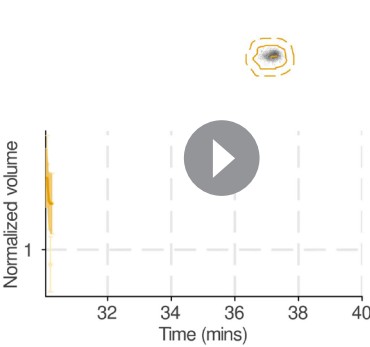

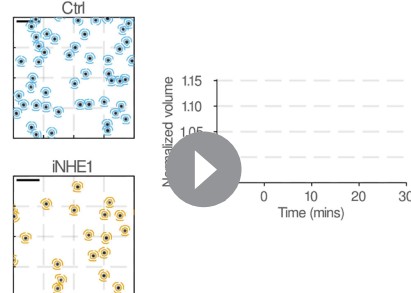

**Video 3.** NHE1-inhibited primary human neutrophils retain the motility-related volume fluctuations but at much lower baseline volumes. Top panel shows a representative cell migrating starting at 30 min following chemoattractant stimulation. The solid gold line is the cell track and the cell's footprint is denoted with a solid gold line while its local background is encapsulated by the dashed gold line. The scale bar is 50 μm. The bottom panel shows the concurrent volume changes with the raw Fluorescence eXclusion Microscopy (FxM) measurements displayed in light gold and the rolling median is shown in gold. The volume is normalized to the median cell volume in the 2 min window prior to stimulation. Data available via Dryad; see https://doi.org/10.7272/Q6NS0S5N.
https://elifesciences.org/articles/90551/figures#video3

**Video 4.** NHE1 inhibition blocks chemoattractant-induced cell swelling and impairs chemokinesis. Left panel shows the individual cells (cyan or gold solid lines) with their individual localities (dashed lines). The top left panel are control cells (cyan) and the bottom left are NHE1 inhibited (gold). The actual Fluorescence eXclusion Microscopy (FxM) signal is visible within the cell footprints. The video is 40 min long with chemoattractant uncaging after the first 10 min. The scale bars are 50 μm. The right panel shows the corresponding median volume of the population in cyan and gold for control and iNHE1, respectively. The corresponding 95% confidence interval is in light cyan or gold. The volume is relative to the average over the 2 min window prior to uncaging. Data available via Dryad; see https://doi.org/10.7272/Q6NS0S5N.
https://elifesciences.org/articles/90551/figures#video4

Since dHL-60 cells exhibit significant basal migration even in the absence of chemoattractant stimulation, they are a non-ideal model for investigating chemoattractant-stimulated migration. Primary human neutrophils, on the other hand, are completely quiescent and non-motile prior to stimulation. We, therefore, sought to replicate our knockout results through pharmacological inhibition of our screen hits in human primary neutrophils. We used BIX (iNHE1), a potent and selective inhibitor of NHE1 (*Huber et al., 2012*), and Duvelisib (iPI3Kγ), a dual PI3Kδ/γ inhibitor (*Winkler et al., 2013*). We compared chemoattractant-stimulated single-cell volume responses in unperturbed, NHE1 inhibited (*Figure 3B*), or PI3Kδ/γ inhibited neutrophils (*Figure 3—figure supplement 1C*). Inhibition of either NHE1 or PI3Kδ/γ prevented chemoattractant-induced swelling in human primary neutrophils. At the single-cell level, the NHE1-inhibited population brackets the initial cell volumes even 30 min post-stimulation, and this is consistent across days and replicates (*Figure 3C*). To orthogonally verify the volume defect, we used a Coulter counter, an electronic particle sizing method, to measure the single-cell volume responses following stimulation in suspension. These experiments confirmed that inhibition of NHE1 blocked cell swelling in suspension as well (*Figure 3—figure supplement 1B*). NHE1-inhibited cells maintained their ability to change shape and spread in response to chemoattractant, though they lagged behind control cells at later time points (*Figure 3D*). In contrast, PI3Kδ/γ inhibition attenuated both the chemoattractant-induced volume change and the shape change (*Figure 3—figure supplement 1B–D*).

Blocking NHE1 activity did not interfere with the spreading-induced volume loss of primary human neutrophils but it prevented the subsequent chemoattractant-induced volume gain. We next sought to determine whether the oscillatory volume fluctuations associated with the motility-cycle were affected by NHE1 inhibition. Performing high temporal resolution imaging of single-cells at later time points (30–50 min) following uncaging revealed that the iNHE1 cells exhibit similar motility-coupled volume changes but at vastly different baselines compared to uninhibited cells (*Figure 3E*; *Video 3*; *Video 4*). Despite the magnitude of the motility-cycle-associated volume fluctuations being similar

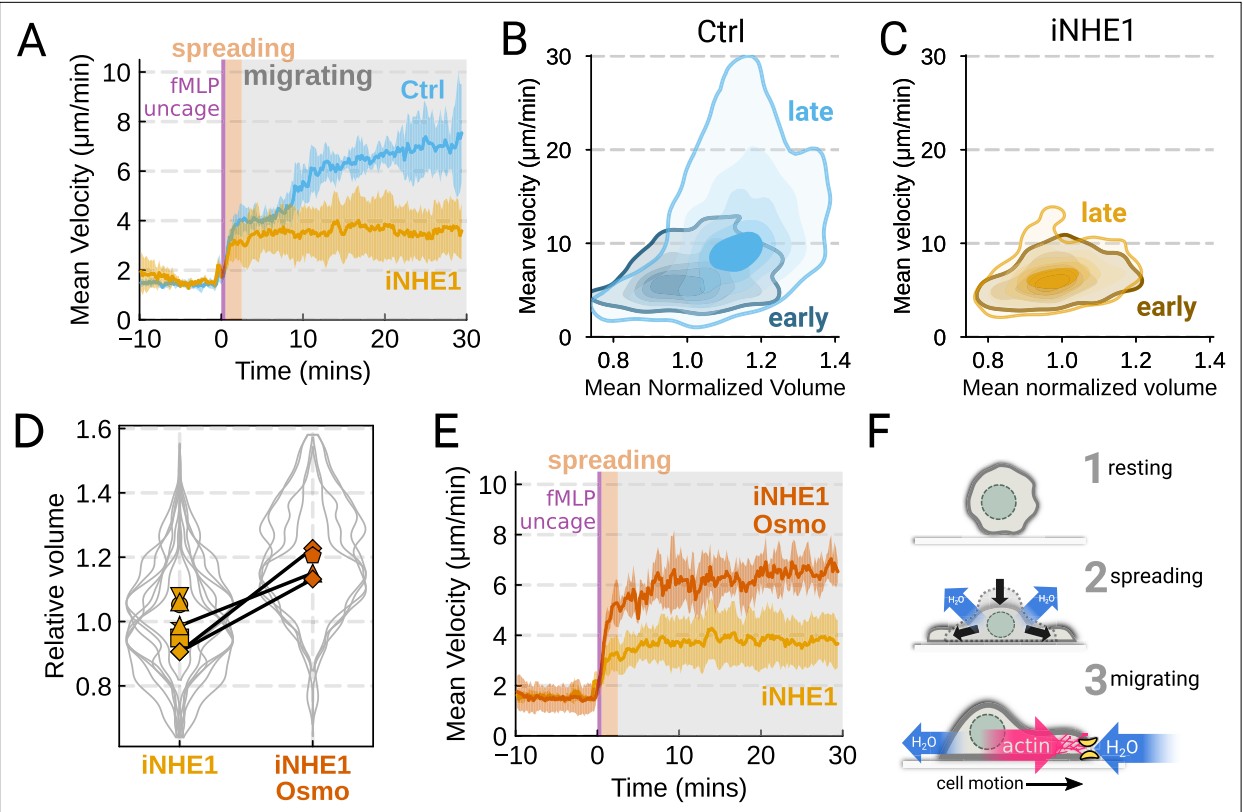

**Figure 4.** The chemoattractant-driven volume gain is necessary and sufficient for rapid cell migration following stimulation (**A**) Comparison of control (blue) or NHE1-inhibited (yellow) primary human neutrophil migration following chemoattractant stimulation. Mean of the per-replicate median cell velocities is shown, with the shaded area indicating standard deviation at each time point. (Ctrl: Volunteer n = 4, iNHE1: Volunteer n = 6). (**B**) Contour plots of the average velocity versus average normalized volume for single unperturbed neutrophils for the initial 10 min window following stimulation (early) and from 20 to 30 min following stimulation (late). (**C**) Contour plots of the average velocity versus average normalized volume for single NHE1-inhibited neutrophils for the initial 10 min window following stimulation (early) and from 20 to 30 min following stimulation (late). (**D**) Dilution of imaging media with 20% water led to a ~15% increase in the median cell volumes of iNHE1 cells (iNHE1 Osmo; red) versus iNHE1 cells in normal media (yellow). Volumes are normalized relative to the median iNHE1 cell volume. This is similar to the magnitude of chemoattractant-induced swelling in control cells. The black lines connect conditions where both conditions were measured for the same volunteer. (iNHE1 Osmo: Volunteer n = 3, 4 total replicates; iNHE1: Volunteer n = 6) (**E**) Testing the ability of cell swelling to rescue the migration defect in NHE1-inhibited neutrophils. Mean of the pre-replicate median cell velocities computed at each time point for NHE1 inhibited cells (yellow) versus mildly hypoosmotically swollen NHE1 inhibited cells (red). Shaded area is the standard deviation at each time point. (iNHE1 Osmo: Volunteer n = 3, 4 total replicates; iNHE1: Volunteer N = 6). See *Video 5* for representative chemokinetic behavior. (**F**) Summary schematic. Cell swelling collaborates with actin polymerization to potentiate chemoattractant-induced cell migration.

The online version of this article includes the following video and figure supplement(s) for figure 4:

**Figure supplement 1.** Additional validation of motility phenotypes.

**Figure 4—video 1.** Inhibiting PI3K signaling interferes with neutrophil chemokinesis.

https://elifesciences.org/articles/90551/figures#fig4video1

(*Figure 3—figure supplement 1E*), the baselines are approximately 20% decreased in NHE1 inhibited versus unperturbed cells following chemoattractant stimulation.

## The chemoattractant-driven volume gain is necessary and sufficient for rapid migration

We next sought to leverage our identified volume regulators to probe the relation between cell swelling and motility. Turning again to the FxM assay, we activated primary human neutrophils by uncaging fMLP and measured the average cell velocity over the population (*Figure 4A*). In the first 10 min following uncaging, both WT and NHE1-inhibited cells exhibited a similar potentiation of migration. However, after 10 min the unperturbed neutrophils continued increasing in velocity, while

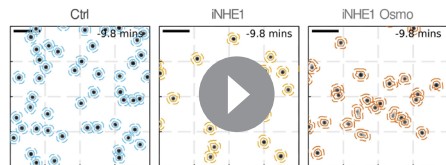

**Video 5.** NHE1-inhibition leads to impaired motility versus control, which is rescued by mild hypoosmotic shock. Each panel is a representative video of control, NHE1-inhibited, and hypoosmotically shocked NHE1-inhibited cells, respectively. Each video is 40 min long and chemoattractant uncaging occurs after 10 min. Scale bars are 50 μm. Data available via Dryad; see https://doi.org/10.7272/Q6NS0S5N.

https://elifesciences.org/articles/90551/figures#video5

the iNHE1 cells plateaued. The WT velocity potentiation is closely correlated with the kinetics of swelling. To visualize the volume-velocity relationship, we plotted the average volume versus average velocity of single WT cells in the first 10 min following uncaging versus 20–30 min post-uncaging (*Figure 4B*). In the early time points following chemoattractant stimulation, control cells operate in a low-volume, low-velocity regime. At later time points following stimulation, control cells operate in a higher-volume high-velocity state. The iNHE1 cells persist in the low-volume, low-velocity state even 20–30 min post-stimulation (*Figure 4C*). To test whether other aspects of chemokinesis are affected in the iNHE1 cells, we also computed the angular persistence of single-cells over 10 micron distance windows and found no difference in between WT and iNHE1 cells (*Figure 4—figure supplement 1F*). This is in contrast to the PI3Kδ/γ-inhibited cells, which failed to increase their velocity following chemoattractant uncaging (*Figure 4—figure supplement 1E*) and fail to migrate persistently post-chemoattractant stimulation (*Figure 4—figure supplement 1F*). The single-cell traces (*Figure 4—figure supplement 1A–D*; *Figure 4—video 1*) show that a large fraction of the PI3Kδ/γ inhibited cells appear to fail to undergo migration following stimulation, offering a potential explanation for the phenotype. In those cells that do start migrating, we observe normal motility-driven volume fluctuations (*Figure 4—figure supplement 1G*) despite defective swelling.

NHE1-inhibited cells are defective in both chemoattractant-induced swelling and rapid migration. To determine if the lack of cell swelling is the basis of their migration defect, we sought to rescue cell swelling for iNHE1 cells through a mild hypoosmotic shock. Diluting the media 20% (v/v) with water led to a 15% increase in the volume of the iNHE1 cells, approximating the magnitude of swelling elicited by fMLP in control cells (*Figure 4D*). Uncaging fMLP-initiated chemokinesis for both iNHE1 and hypo-iNHE1 cells, but the hypoosmotically shocked cells continued accelerating for longer and reached greater sustained velocities (*Figure 4E*; *Video 5*). Intriguingly, the hypoosmotically shocked cells are precocious in their rapid motility. This is expected, since these cells are pre-swollen prior to stimulation, whereas control cells take longer to reach the high-volume high-velocity state following chemoattractant stimulation. Our data suggest that the water influx following chemoattractant stimulation plays an important role in the potentiation of neutrophil migration (*Figure 4F*).

## Discussion

Rapid migration is key to the innate immune function of neutrophils (*Lämmermann et al., 2013*). Here, we show that human primary neutrophils actively increase their cell volumes when stimulated with chemoattractant, and this correlates with their potentiation of movement (*Figure 1*). We find that the chemoattractant-induced swelling occurs independently of cell spreading and motility (*Figure 1—figure supplement 2*), enabling us to isolate and study the specific role of the swelling on migration. We performed an unbiased genome-wide screen to identify the molecular components of chemoattractant-induced cell swelling (*Figure 2D–F*). While AE2 and particularly, NHE1, have been investigated in previous studies (*Li et al., 2021*; *Rosengren et al., 1994*; *Ritter et al., 1998*; *Denker and Barber, 2002*; *Frantz et al., 2007*; *Zhang et al., 2022*), our work systematically identifies the dominant players in a larger network that contribute to the swelling response. Buoyant density screening was used with great success by Schekman and colleagues to elucidate the secretory pathway in yeast (*Novick et al., 1980*; *Novick and Schekman, 1979*). Here, we use the sensitivity of this assay combined with the power of modern forward genetics to uncover the mechanistic basis of how cells actively manipulate their volume to enhance migration. Similarly to how fungi and plant

cells balance cell wall mechanics with hydraulics to control cell expansion (*Dumais, 2021*), animal cells could use water flows to enhance cytoskeletal dynamics during motility.

Our work implicates both NHE1 and AE2 in cell swelling, as knockout of either completely ablates chemoattractant-induced swelling in dHL-60 cells (*Figure 3A*). Similarly, knockouts of CA2 and PI3Kγ reduced the degree of swelling (*Figure 3—figure supplement 1A*). We then confirmed our dHL-60 results via pharmacological inhibition in human primary neutrophils. We found that PI3Kδ/γ inhibition interfered with both swelling and chemokinesis in primary human neutrophils (*Figure 3—figure supplement 1B–D*; *Figure 4—figure supplement 1D–F*). Additionally, we verified the necessity of NHE1 in the chemoattractant-induced swelling response (*Figure 3*). NHE1-inhibited cells showed a defect in both motility and chemoattractant-induced swelling (*Figure 4A–C*). At longer time points, NHE1-inhibited cells fail to continue their migration acceleration compared to uninhibited cells. This lack of migration potentiation in NHE1-inhibited cells can be explained by their defective cell swelling, as exogenous swelling via hypoosmotic shock rescues the velocity defect (*Figure 4E*).

How might swelling contribute to rapid cell migration? Given that neutrophils are approximately 65% water (*Figure 2—figure supplement 1F*), the 15% increase in cell volume corresponds to almost a 25% increase in the water content of the cell after 20 min. This change could affect global biophysical parameters by decreasing cytoplasmic viscosity or increasing the diffusion of biochemical or cytoskeletal regulators of movement. Alternatively, local net water influx could collaborate with the actin polymerization machinery in facilitating the extension of the plasma membrane (*Mitchison et al., 2008*; *García-Arcos et al., 2022*). The regulatory volume components identified here are ubiquitously expressed, so it is possible that they also facilitate chemoattractant-induced migration in other contexts. NHE1-dependent swelling has been observed in dendritic cells responding to LPS (*Rotte et al., 2010*), and NHE1 inhibition slows microglial chemotaxis (*Shi et al., 2013*). NHE1's role in migration and metastasis is well-established (*Stroka et al., 2014*; *Zhang et al., 2022*; *Klein et al., 2000*; *Denker and Barber, 2002*), but whether it plays an active role or merely passively maintains the cytoplasmic pH is still debated (*Hayashi et al., 2008*). Our experiments indicate that NHE1's active role in the cell swelling program is critical to the potentiation of migration. The other constituents of our chemoattractant-induced cell swelling program have also been implicated in cell migration. AE2 plays a role in murine osteoclast spreading and migration (*Coury et al., 2013*). Similarly, carbonic anhydrases have been implicated in enhancing NHE1 activity (*Li et al., 2006*) and facilitating migration (*Svastova et al., 2012*), and PI3K isoforms have a well-appreciated role in migration (*Ferguson et al., 2007*; *Devreotes and Horwitz, 2015*). Systematic investigation of this chemoattractant-induced cell swelling network could reveal a general role for water influx in potentiating animal cell migration.

## Materials and methods
### Medias and inhibitors

For all imaging experiments, imaging media was made with Phenol Red-free Leibovitz's L-15 media (Gibco #21083027) supplemented with 10% (v/v) heat-inactivated fetal bovine serum (Gibco) and filtered with a 0.22 um Steriflip filter (MilliporeSigma #SCGP00525). Imaging media was always prepared fresh on the same day of imaging. For washing FxM chips, 0.2% endotoxin, and fatty acid-free Bovine Serum Albumin (BSA) (Sigma #A8806) was dissolved in L15 via pulse centrifugation. The mix was then filtered with a Steriflip filter before further use. For all density experiments, divalent-free mHBSS media was prepared as in *Houk et al., 2012*. In short, 150 mM NaCl, 4 mM KCl, 10 mg/mL glucose, and 20 mM HEPES were dissolved in Milli-Q (Millipore) water and the pH-adjusted to 7.2 with 1 M NaOH. The osmolarity was verified to be 315 mOsm/kg on a micro-osmometer (Fiske Model 210). Culturing media (R10) was made from RPMI 1640 media (Gibco #11875093) supplemented with 25 mM HEPES and L-glutamine supplemented with 10% (v/v) heat inactivated fetal bovine serum (Gibco). The NHE1 inhibitor, BIX (Tocris #5512), was dissolved in dry DMSO to a final concentration of 25 mM and stored at –20 °C in single use aliquots that were diluted in imaging media on the day of the experiment. All iNHE1 experiments used BIX at a 5 uM final concentration. Similarly, Latrunculin-B (Sigma #428020) was stored at 10 mM in DMSO and used at 1 uM final. For PI3Kδ/γ inhibition, Duvelisib (MedChemExpress #HY-17044) was stored at 10 mM in DMSO and used at a final concentration of 1 uM.

## Human primary neutrophil isolation and drug treatment

All blood specimens from patients were obtained with informed consent according to the institutional review board-approved study protocol at the University of California - San Francisco (Study #21–35147), see *Supplementary file 1B* for demographic information. Fresh samples of peripheral blood from healthy adult volunteers were collected via a 23-gauge butterfly needle collection set (BD #23-021-022) into 10 ml Vacutainer EDTA tubes (BD #366643). Blood was kept on a shaker at minimum setting and utilized within 2 hr of the draw. Neutrophils were isolated using the EasySep Direct Human Neutrophil Isolation Kit (STEMCELL Tech #19666) with the BigEasy magnet (STEMCELL Tech #18001) according to the manufacturer's protocol.

Isolated neutrophils were spun down at 200 × g for 5 min and resuspended in a dye media consisting of imaging media containing 5 µg/ml Hoechst 3334 (Invitrogen #H3570) and 0.25 uM Calcein Red-Orange AM (Invitrogen #C34851). This cell suspension was incubated at room temperature in the dark for 15 min, and then the cells were spun down at 200 g for 5 min. The dye medium was aspirated and replaced with R10 to achieve a final cell density at or below $1 \times 10^6$ cells/mL. Purified neutrophils were then kept in polystyrene T25 flasks (Corning) at 37 °C in a 5% CO2 environment until imaging. Cells were used ~5–8 hr post-isolation.

## Cell culture

Short tandem repeat authenticated HL-60 cells (*Saha et al., 2023*) were maintained in R10 media at 5% CO2 and 37 °C and at a concentration of 0.2–1 million/mL by passaging them every 2–3 days. 5 days prior to experiments, HL-60s were differentiated into a neutrophil-like state by taking an aliquot of cells in their culturing medium and supplementing with an equal volume of Nutridoma-CS (Roche #11363743001) and DMSO diluted in RPMI such that the final concentrations were 0.2 million/mL HL-60 cells, 2% (v/v) Nutridoma-CS, 1.3% (v/v) DMSO, 5% (v/v) FBS in RPMI. After 5 days at 37 °C/5% CO2, we observed robust expression of terminal differentiation markers like FPR1 as reported previously (*Rincón et al., 2018*).

Lenti-X HEK-293Ts (Takara) were used for lentivirus production and maintained at below 80% confluency in DMEM supplemented with 10% (v/v) heat-inactivated fetal bovine serum. These cells were also maintained at 5% CO2 and 37 °C. All cell lines were routinely monitored for mycoplasma contamination using standard mycoplasma monitoring kits (Lonza).

## FxM single-cell volume measurements

FxM microfluidic chips were prepared as previously described *Zlotek-Zlotkiewicz et al., 2015*; *Cadart et al., 2017* using a custom mold generously provided by the Piel lab. Briefly, 10:1 (w/w) PDMS elastomer base and crosslinker (Momentive #RTV615-1P) were thoroughly mixed, poured into the FxM mold, and degassed under a vacuum for 1 hr. The PDMS was then baked at 80 °C for 2 hr and removed from the mold. The day prior to experiments, the molded PDMS was cut with a scalpel to form three lane 'chips' and the inlet and outlet holes were created using a 0.5 mm punch. The chips and 35 mm glass-bottomed dishes (Willco Wells #HBST-3522) were then plasma cleaned for 30 s, and chips were gently pressed down onto the glass to form a watertight seal. A good seal was verified visually by the refractive index change upon glass/PDMS contact. The chips were then baked at 80 °C for 10 min to ensure thorough bonding. The chips were then quickly coated with 100 µg/mL human fibronectin (Sigma #SLCL0793) diluted in PBS and injected using a pipette tip. Coating was allowed to proceed for 30 min at RT before the chamber was flushed with imaging media. The chips were then submerged in PBS and allowed to incubate with L15 +0.2% BSA overnight at 4 °C.

On the day of the experiment, pre-prepared microfluidic chips were allowed to warm up at RT. Human primary neutrophils were gently pipetted up and down to resuspend if they had settled and spun down at 200 × g for 4 min. The cell pellet was very slowly resuspended in imaging media to achieve a cellular concentration of 60 million per mL. The cells were allowed to equilibrate for 30 min at RT. This step helped reduce gas production by the cells that caused problematic laminar flows in our chips. The lanes of the chip were flushed with the corresponding final media. The cells were gently mixed with a 2 x solution such that the final concentrations were 0.5 mg/mL Alexa Fluor 647-tagged 10,000 MW dextran (Invitrogen #D22914), 200 nM caged fMLP (NEP), and 30 million per mL cells in imaging media. This mixture was then slowly pulled into the chamber using a partially depressed pipette tip to minimize the shear forces on the cells, as these are known to affect neutrophil

response to fMLP (*Mitchell and King, 2012*). Once loading was complete, the entire chamber was submerged in imaging media to stop all flows and allowed to warm up to 37 °C. Experiments were started promptly 20 min post-submersion.

## Suspension cell volume measurements

Suspension cell volume measurements were performed as in *Graziano et al., 2019*. Briefly, human primary neutrophils were spun out of culture media at 200 × g for 4 min and gently resuspended in mHBSS. They were then diluted to 20,000 cells/mL in warm 15 mL of mHBSS in Accuvettes (Beckman-Coulter). The cells were incubated at 37 °C for 5 min, and then either a DMSO blank or the indicated amount of drug was added to the correct final concentration. The cells were again incubated for 5 min at 37 °C. They were then quickly transported to the Multisizer Z2 instrument (Beckman-Coulter) at RT. Three time points were taken to set a baseline, and then fMLP (Sigma) was added to a final concentration of 20 nM, and the Accuvette was inverted to mix. Then 0.5mL samples were taken continuously every minute using a 100 um diameter aperture with a current of 0.707 mA, a gain of 64, a pre-amp gain of 179.20, a calibration factor (Kd) of 59.41 and a resolution of 256 bits. 5000–10,000 cells were sampled per time point and the medians of the population was extracted using our software available on GitHub (copy archived at *Nagy, 2024a*).

## Buoyant density measurements

Buoyant density measurements were done by pre-pouring gradients, layering dHL-60s on top, centrifuging, fractionating, and then imaging to count cells. First, solutions were made with either 32.6% or 57% (v/v) Percoll (Sigma) with 10% (v/v) 10 x divalent-free mHBSS and diluted with ultrapure water, making a low density solution (LDS), and high density solution (HDS), respectively. The refractive index of both solutions was determined with a MA871 refractometer (Milwaukee Instruments) as 1.3419 and 1.3467, respectively. Given that the density is linearly related to the refractive index (*Figure 2—figure supplement 1A*) the solutions have densities of 1.045 g/mL and 1.074 g/mL, respectively. For the chemoattractant-condition, 20 nM fMLP (Sigma) was added. A linear gradient mixer was attached to an Auto Densi-Flow (Labconco) gradient fractionator and used to dispense gradients into 14 mL round bottom tubes (Falcon #352041).

For each gradient, approximately 5 million dHL-60s were spun down at 200 × g for 4 min and resuspended in 1 mL of 1 x mHBSS. The cells were then labeled with 0.5 μM Calcein-AM (Invitrogen) for 5 min then spun down and resuspended in LDS. For mixed populations, the two cell types were spun down and labeled separately with either Calcein-AM or Calcein Red-Orange-AM and then mixed together. The cells were layered gently on top of the gradient and spun at 250 × g for 1 hr. Neutrophils display homotypic aggregation when activated during centrifugation (*Simon et al., 1990*) so we used a divalent-free media and very long centrifugation times optimized for separation at low centrifugation speeds (*Figure 2—figure supplement 1C*).

After centrifugation, the cells were fractionated into a 96-well using the Auto Densi-Flow in 'remove' modality and a homemade fractionator. Six to seven wells were taken, and their refractive index was measured using the refractometer to align the gradients and verify linearity (*Figure 2—figure supplement 1B*). A 2 x volume of blank media was added to each well to reduce the density, and then the plates were spun in the centrifuge at 250 × g to assist in the settling of the cells on the glass. The plates were then imaged using confocal microscopy to determine the number of cells in each well. For mixed-population experiments, dual-color imaging was done to determine the cell count of each population.

## CRISPR genome-wide screen on buoyant density

Lenti-X 293Ts (Takara) were transfected with the Guide-it library (Takara) according to the manufacturer's instructions and concentrated ~100 x using the Lenti-X concentrator kit (Takara) and stored at –80 °C until needed. Human codon-optimized *S. pyogenes* Cas9-tagBFP expressing HL-60 cells (*Graziano et al., 2019*) were transduced by spinoculating the cells on Retronectin-coated (Takara) non-TC treated six-well plates (Falcon #351146). Briefly, each well was coated with 20 μg/mL Retronectin stock solution diluted in DPBS for 2 hr and then blocked with 2% BSA (w/v) in PBS for 30 min and washed with PBS. 2 mL of 1 million/mL Cas9-BFP HL-60s were added to each well of four plates (48 million cells total) and 30 uL of concentrated Guide-it library lentivirus was added to each

well. Using Lenti GoStix (Takara), we estimated that this corresponds to $8 \times 10^6$ IFUs per well. The plates were spun at $1000 \times g$ for 30 mins, and then 1 mL of R10 was added gently. This was followed by another dilution with 2 mL of the same media after 24 hr. 48 hr post spinoculation, the cells were spun down at $200 \times g$ and resuspended in R10. The cells were then sorted for mCherry-positive cells (cutoff set at 99.9th percentile of the untransduced cell population's mCherry signal) using a FACSAria 3 cell sorter (BD). We observed 4% of cells with a mCherry positive signal, equivalent to a MOI of 0.04. This gives a minimum coverage of 6–24 x of the library at transduction; post-sequencing Monte Carlo simulations suggest a minimum coverage of 12 x. After sorting, the cells were selected using 175 µg/mL hygromycin and kept in log-phase growth with regular supplementation with fresh media for 7 days, after which ~95% of the population were mCherry positive.

5 days prior to screening day, the cells were differentiated into neutrophil-like cells as described in the 'Cell Culture' section. The buoyant density assay was performed as described in the 'Buoyant Density Measurements' section with 6 million cells per tube split across 6 tubes, corresponding to 36 million cells or ~450 x coverage of the library. The cells were layered on top of the gradients containing 20 nM fMLP with or without 1 uM Latrunculin-B, and then each tube was fractionated into 48 wells, which were combined into 3 separate bins such that they each contained approximately one-third of the population (*Figure 2—figure supplement 1D*). The bins from each tube belonging to the same sample were combined, and then the cells were spun down, and the pellets were flash-frozen to store for further processing.

The genomic DNA was extracted using the QIAamp gDNA kit (Qiagen) according to the manufacturer's instructions. The guides were then PCR amplified for 26 cycles using the Ex Taq polymerase (Takara) protocol with the P5 forward primer mix (Takara) and a unique reverse P7 primer for each condition. The specificity and quality of amplification for each sample was validated using a TapeStation 4200 (Agilent), and the precise DNA concentration was determined using a Qubit fluorometer (Invitrogen) according to manufacturers' instructions. The amplified DNA was then pooled to a 10 nM final concentration followed by a 5% PhiX (Illumina) spike and sequenced in a PE100 run on a HiSeq 4000 sequencer (Illumina) at the UCSF sequencing core.

To verify that we can detect the functioning of Cas9, we assayed for the depletion of guides targeting previously published essential genes (*Wang et al., 2015*; *Evers et al., 2016*). We used MAGeCK (*Li et al., 2014*) to compute the log fold change between the known frequencies of the guides in the library (Takara) and the actual observed frequencies of the guides (computed by pooling all bins together). The cumulative distribution of the essential gene ranks were compared to a randomly shuffling of those same genes to demonstrate that the essential genes were highly depleted from the population, as expected (*Figure 2—figure supplement 1E*).

Similarly, to determine which genes were involved in the chemoattractant-induced density change, we used MAGeCK to compute the false discovery rate and log fold change between the first and second bins vs the third bin (*Figure 2—figure supplement 1D*). The third bin was the most dense bin, so genes over-represented in this bin versus the other two were likely interfering with the swelling process. We pooled the samples with and without 1 uM Latrunculin-B together to improve our sensitivity as the swelling is not dependent on a polymerized cytoskeleton (*Figure 1—figure supplement 2*). The combined fold change for each gene was then computed by taking the MAGeCK 'alphamedian' log fold change (either positive or negative) that was most divergent. Using this fold change and the false discovery rate, we identified the genes that are most likely involved with the chemoattractant-induced swelling (*Figure 2E*, *Supplementary file 1A*).

## Single gene knockout line generation with CRISPR/Cas9

Single gene knockouts were generated and validated using wildtype HL-60s expressing human codon-optimized *S. pyogenes* Cas9-tagBFP cells as the base line as previously described (*Graziano et al., 2019*). The two best performing guides from the genome-wide screen (*Supplementary file 1C*) were selected and synthesized (IDT) and then cloned into the pLVXS-sgRNA-mCherry-hyg Vector (Takara) following the manufacturer's instructions. Lentivirus was then produced as previously described (*Graziano et al., 2019*). Briefly, LX-293T cells (Takara) were seeded into 6-well plates and grown till 80% confluence was reached. 1.5 µg of the guide vector (from above) was mixed with 0.167 µg vesicular stomatitis virus-G vector and 1.2 µg cytomegalovirus 8.91 vector. This mixture was incubated with the TransIT-293 Transfection Reagent (Mirus Bio) and used to transfect the 293T cells following

the manufacturer's instructions. The cells were grown for 72 hr post-transfection and the virus was concentrated ~40 fold from the supernatant using the Lenti-X concentrator kit (Takara) per the manufacturer's protocol. Concentrated virus stocks were stored at –80 °C until needed. For transduction, virus stocks were thawed and added to 300,000 cells in R10 in the presence of polybrene (8 ug/mL) and incubated for 24 hr. Afterwards, the cells were washed twice with R10 to remove any remaining viral particles and sorted for mCherry-positivity on a FACSAria 3 cell sorter (BD). The heterogeneous population was then assayed for successful editing by sequencing the genomic DNA flanking the Cas9 cut site. Clonal populations were then isolated by seeding dual BFP and mCherry-positive cells into a 96-well plate such that only one cell was deposited in each well using a FACSAria Fusion (BD). The cells were then allowed to grow up and clonality was verified by genomic DNA sequencing of the cut site as previously described (*Graziano et al., 2017*).

## Microscopy hardware

FxM and buoyant density experiments were performed on an inverted Eclipse TI microscope (Nikon) with a Borealis beam conditioning unit (Andor), and light was collected on an air-cooled iXon 888 Ultra EM-CCD (Andor). A 20 x Plan Apochromat NA 0.75 objective (Nikon) was used for FxM, and a 10 x Plan Apochromat (Nikon) was used for density experiments. Light sources include a Stradus Versalace 405, 488, 561, 647 nm laser line system (Vortran Laser Technologies) and a Sutter Lambda LS xenon-arc lamp used for FxM. Microscopy hardware was controlled with a TriggerScope 4 (Advanced Research Consulting) via MicroManager (*Edelstein et al., 2014*).

UV light for uncaging was delivered via a 365 nm Lambda FLED (Sutter) launched into a Lambda OBC and delivered via the condenser with all mobile optical elements removed and all apertures wide open. Before every experiment, the wattage was measured using a light meter (Thor Labs). The LED was controlled via a custom MicroManager script.

## Data analysis

FxM images were analyzed using a custom pipeline (*Figure 1—figure supplement 1A*) implemented in the Julia language (*Bezanson et al., 2017*) available on GitHub (copy archived at *Nagy, 2024b*). Briefly, the raw images were denoised with a patch-based algorithm (*Boulanger et al., 2010*) and then the edges were enhanced using a Scharr kernel. The magnitude of the edge values is log normally distributed, and we empirically determined that calling the edge of the cell at one standard deviation above the mean gave low noise and the maximum signal (*Figure 1—figure supplement 1C*). Flood-filling the areas encapsulated by these edges gave a binary mask of the foreground, i.e., space occupied by the cells.

Independently, instance segmentation was performed using a custom Cellpose model trained using the human-in-the-loop feature (*Pachitariu and Stringer, 2022*) on the raw nuclear and cytoplasmic channels. Trackpy was used to link the Cellpose segmented instances together in time (*Allan et al., 2021*) and these were then used to nucleate a watershed algorithm on the binary mask of the foreground and separate the foreground into the individual tracked cells. Next, the raw image was flatfield corrected by fitting a multiquadratic function at sinusoidally placed locations (avoiding the locations of cells or pillars) in the raw image which gave the densest sampling at the edges. Given that the chambers have flat ceilings supported by pillars (*Zlotek-Zlotkiewicz et al., 2015*), the background signal can be used to compute the per-time point flatfield. After subtracting the darkfield image from both the raw FxM signal and the interpolant, the raw signal was divided by the interpolant giving extremely uniform homogeneous signal across the FOV.

Next, the local background was computed for each cell which we defined as the region 2–10 pixels away (1.3–6.5 microns) from the cell borders while avoiding other cells. The median of this local background was used to compute the counterfactual of what the signal would have been if the cell was not there by multiplying by the pixel area of the cell $A$ (*Figure 1B*, inset). The measured signal over the area of the cell was then subtracted from that value to give the volume excluded by the cell according to Equation 2 from *Cadart et al., 2017*:

$$V_{cell} = \sum_{x,y \in A} \frac{I_{max} - I_{cell}(x, y)}{\alpha} \tag{1}$$

where $I_{max}$ is the median signal of the locality, i.e., the maximum signal if the cell was not there. $A$ is the cell footprint area and $I_{cell}$ is the signal at each point $x, y \in A$. To convert this to absolute volume measurements we divide by $\alpha$ which is the fluorescence as a function of object height:

$$\alpha = (I_{max} - I_{min})/h_{chamber} \tag{2}$$

Where the minimum signal $I_{min}$ is the signal at the pillars that support the chamber's roof and $h_{chamber}$ is the height of the chamber in microns. As discussed in *Cadart et al., 2017*, while the per-pixel heights might not be accurate due to light scatter, segmenting a slightly larger area than the cell footprint (*Figure 1—figure supplement 1C*) captures any scattered signal and yields accurate whole cell volumes.

For velocity measurements, cell tracks were analyzed using the following equation to compute the velocity at frame $i$ over a window $\tau$ given the $x$ and $y$ coordinates in microns and time $t$ in seconds:

$$\nu_i = \frac{\sqrt{(x_{i+\tau} - x_{i-\tau})^2 + (y_{i+\tau} - y_{t-\tau})^2}}{t_{i+\tau} - t_{i-\tau}} \tag{3}$$

For all plots in this paper a $\tau$ of 3 corresponding to approximately a 1 min window was used, but similar results were obtained at other values of $\tau$.

To compute angular alignment, we first computed the cumulative distance traveled by each cell at each time-point $i$. We then computed the angle $\angle$ between each position $B_i$ and positions occupied by the cell prior $A_i$ and subsequent $C_i$ separated by 10 microns of distance traveled by the cell:

$$\text{angular alignment} \angle_i = \text{atan2} \left( \frac{\|\overrightarrow{B_iA_i} \times \overrightarrow{B_iC_i}\|}{\overrightarrow{B_iA_i} \cdot \overrightarrow{B_iC_i}} \right) \tag{4}$$

The angle $\angle_i$ was then re-scaled to be between –1 (complete reversal) and 1 (completely straight) by dividing by $\pi$. The results were robust to choice of either distance or temporal offset of $A$ and $C$ from $B$ (data not shown). Finally, to compute the population angular alignment for each time-point, we used the median of $\angle_i$. As expected, the median angular alignment prior to activation is close to 0 as the cells display Brownian motion, but then increases as cells start migrating persistently (*Figure 4—figure supplement 1F*).

## Acknowledgements

We thank Andrea Eastes, Ram Adar, and Kate Cavanaugh for a critical reading of the manuscript, and all members of the Weiner lab for their support and discussions. We also thank Larisa Venkova and Matthieu Piel for the custom shallow FxM mold used in this work. This work was supported by a NSF Graduate Research Fellowship (TLN), UCSF Discovery Fellowship (TLN), an American Heart Association Predoctoral Fellowship (ES), National Institutes of Health grant GM118167 (ODW), National Science Foundation/Biotechnology and Biological Sciences Research Council grant 2019598 (ODW), the National Science Foundation Center for Cellular Construction (DBI- 1548297, ODW), and a Novo Nordisk Foundation grant for the Center for Geometrically Engineered Cellular Systems (NNF17OC0028176, ODW). Sequencing was performed at the UCSF CAT, supported by UCSF PBBR, RRP IMIA, and NIH 1S10OD028511-01 grants.

## Additional information

### Funding

| Funder | Grant reference number | Author |
| --- | --- | --- |
| National Science Foundation | Graduate Research Fellowship | Tamas L Nagy |
| University of California, San Francisco | Discovery Fellowship | Tamas L Nagy |

| Funder | Grant reference number | Author |
|---|---|---|
| American Heart Association | Predoctoral Fellowship | Evelyn Strickland |
| National Institutes of Health | GM118167 | Orion D Weiner |
| National Science Foundation | BBSRC grant 2019598 | Orion D Weiner |
| National Science Foundation | Center for Cellular Construction DBI-1548297 | Orion D Weiner |
| Novo Nordisk | Center for Geometrically Engineered Cellular Systems NNF17OC0028176 | Orion D Weiner |

The funders had no role in study design, data collection and interpretation, or the decision to submit the work for publication.

### Author contributions

Tamas L Nagy, Conceptualization, Resources, Data curation, Software, Formal analysis, Validation, Investigation, Visualization, Methodology, Writing – original draft, Writing – review and editing; Evelyn Strickland, Resources, Methodology, Writing – review and editing; Orion D Weiner, Conceptualization, Supervision, Funding acquisition, Methodology, Project administration, Writing – review and editing

### Author ORCIDs

Tamas L Nagy ⬛ https://orcid.org/0000-0001-6080-9782
Orion D Weiner ⬛ https://orcid.org/0000-0002-1778-6543

### Ethics

All blood specimens from patients were obtained with informed consent according to the institutional review board-approved study protocol at the University of California - San Francisco (Study #21-35147), see Table S2 for demographic information.

Reviewer #1 (Public review): https://doi.org/10.7554/eLife.90551.3.sa1
Reviewer #2 (Public review): https://doi.org/10.7554/eLife.90551.3.sa2
Author response https://doi.org/10.7554/eLife.90551.3.sa3

## Additional files

### Supplementary files

• MDAR checklist

• Supplementary file 1. Extended data tables for CRISPR screen and volunteer demographic information. (A) Chemoattractant-induced swelling genome-wide CRISPR KO screen hits. Rankings determined using MAGeCK *Li et al., 2014*. Fold change is the median fold enrichment in the dense bin versus the other two bins of the functional guides. FDR is the false discovery rate of each gene given the distribution of negative control guides in the library. See Methods section for details. (B) Volunteer Demographic Information. Demographic information collected for the volunteer donors according to the Institutional Review Board-approved study protocol at the University of California - San Francisco (Study #21-35147). (C) Guides used to make single gene knockouts in HL-60s. The two highest performing guides from the genome-wide screen were chosen to make single gene knockouts in the HL-60 cell line. See Methods for details.

### Data availability

All Fluorescence Exclusion Microscopy datasets and Coulter datasets are deposited with Dryad. Both raw and processed datasets are included due to the use of a licensing-encumbered denoising algorithm (*Boulanger et al., 2010*) during processing. The fxm_uncaging_datasets.csv file on Dryad details each dataset and includes rationale if a dataset was excluded from the analysis. The primary reason for FxM dataset exclusion was the occasional presence of strong laminar flows, preventing cell adhesion and migration, due to technical limitations with the design of the microfluidic chip. The

code used to generate the figures is available on Zenodo. The best way to learn how to interact with the data is to explore the code used to generate the figures in the paper. To facilitate this, the code is available as a mini-site (accessible via the above Zenodo link) which is built automatically on GithHub Actions to validate reproducibility and portability.

The following dataset was generated:

| Author(s) | Year | Dataset title | Dataset URL | Database and Identifier |
|-----------|------|---------------|-------------|------------------------|
| Nagy T, Strickland E, Weiner O | 2023 | Data from: Neutrophils actively swell to potentiate rapid migration | https://doi.org/10.7272/Q6NS0S5N | Dryad Digital Repository, 10.7272/Q6NS0S5N |

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
