## [Editor Report · eLife assessment]

This **fundamental** study significantly advances our understanding of the role of water influx and swelling on neutrophil migration in response to chemoattractant. The evidence supporting the conclusions, based on a genome-wide CRISPR screen and high quality cellular observations, is **compelling**. This paper will be of interest to cell biologists and biophysicists working on cell migration.

---

## [Referee Report · Reviewer #1 (Public review)]

Summary:

The authors use innovative CRISPRi method to uncover regulators of cell density and volume in neutrophils. The results show that cells require NHE activity during chemoattractant-driven cell migration. Before migration occurs, cells also undergo a rapid cell volume increase. These results indicate that water flux, driven by ion channels, appears to play a central role in neutrophil migration. The paper is very well written and clear. The revised version has addressed all of my questions.

---

## [Referee Report · Reviewer #2 (Public review)]

Nagy et al investigated the role of volume increase and swelling in neutrophils in response to the chemoattractant. Authors show that following chemoattractant response cells lose their volume slightly owing to the cell spreading phase and then have a relatively rapid increase in the cell volume that is concomitant with cell migration. Authors performed an impressive genome-wide CRISPR screen and buoyant density assay to identify the regulators of neutrophil swelling. This assay showed that stimulating cells with chemoattractant fMLP lead to an increase in the cell volume that was abrogated with the FPR1 receptor knockout. The screen revealed a cascade that could potentially be involved cell swelling including NHE1 (sodium-proton antiporter) and PI3K. NHE1 and PI3K is required for chemoattractant-induced swelling in human primary neutrophils. Authors also suggest slightly different functions of NHE1 and PI3K activity where PI3K is also required for maintain chemoattractant-induced cell shape changes. Authors convincingly show that chemoattractant induced cell swelling is linked to cell migration and NHE1 is required for swelling at the later stages of swelling since the cells at the early point work on low-volume and low-velocity regime. Interesting authors also show that lack of swelling in NHE1 inhibited cells could be rescued by mild hypo-osmotic swelling strengthening the argument that water influx followed chemoattractant stimulation is important for potentiation for migration.

The conclusions of this paper are mostly well supported by data and is pretty convincing

---

## [Author Response]

The following is the authors’ response to the original reviews.

**Reviewer #1 (Public Review):**
Summary:The authors use the innovative CRISPRi method to uncover regulators of cell density and volume in neutrophils. The results show that cells require NHE activity during chemoattractant-driven cell migration. Before migration occurs, cells also undergo a rapid cell volume increase. These results indicate that water flux, driven by ion channels, appears to play a central role in neutrophil migration. The paper is very well written and clear. I suggest adding some discussion about the role of actin in the process, but this is not essential.StrengthsThe novel use of CRIPSPi to uncover cell density regulators is very novel. Some of the uncovered molecules were known before, e.g. discussed in Li & Sun, Frontiers in Cell and Developmental Biology, 2021. Others are more interesting, for example PI3K-gamma. The use of caged fMLP is also nice.

We thank the reviewer for their positive appraisal of our work and have pursued their suggestions for improving our paper in this revision.

WeaknessesOne area of investigation that seems to be absent is mentioned in the introduction. I.e., actin is expected to play a role in regulating cell volume increase. Did the authors perform any experiments with LatA? What was seen there? Do cells still migrate with LatA, or is a different interplay seen? The role of PI3K is interesting, and maybe somewhat related to actin. But this may be a different line of inquiry for the future.

We agree that we could have done a better job explicitly investigating the role of actin dynamics in volume changes. Towards this end, by using Latrunculin B to depolymerize actin, we find that the volume increase in suspension is not affected (Figure 1 – supplemental figure 2A). In our FxM single cell volume measurements of adherent cells, we similarly observed unhindered swelling following latrunculin treatment. These data indicate that actin is dispensable for chemoattractant-induced cell swelling (Figure 1 – supplemental figure 2B) . There was a minor apparent reduction in the final volume reached with the Latrunculin-treated cells as measured by FxM, but this likely reflects minor uptake of the excluded dye following Latrunculin treatment rather than an actual change in final volume. This conclusion is reinforced by the change in 2D footprint area being well modeled by the 2D projection of an isotropically expanding sphere (Figure 1 – supplemental figure 2C) . Latrunculin treatment completely abolishes migration, as is expected for unconfined migration on fibronectin (Figure 1 – supplemental figure 2D-E) . The second Reviewer also wanted us to dig deeper on the role of PI3K-gamma, so we expanded our analysis of this hit (Figure 3 – supplemental figure 1B-D; Figure 4 – supplemental figure 1D-G) .

**Author response image 1. sa3fig1:** Chemoattractant-induced swelling, but not motility, is independent of actin polymerization. (**A**) Human primary neutrophils were incubated with DMSO or Latrunculin B, activated with 20 nM fMLP, and then volume responses were measured using electronic sizing via a Coulter counter. Latrunculin treatment did not alter cell swelling, indicating that actin polymerization is dispensable for the chemoattractant-induced volume increase. (**B**) Similar results were obtained using the FxM assay, showing that Latrunculin-treated cells are capable of swelling after stimulation. (**C**) The Latrunculin-treated cells also increase their footprints, albeit less so than control cells, but this is within the range of what would be expected for this degree of chemoattractant-induced volume increase (modeled by a sphere expanding an equivalent volume). (**D**) Single cell tracks of primary human neutrophils responding to acute chemoattractant stimulation. Both panels show 15 minutes of tracks with the tracks prior (left) and the 15 minutes post (right) uncaging the chemoattractant. The scale bar is 50 microns. The top panels show the large increase in motility displayed by control cells, while the Latrunculin-treated cells (bottom panels) fail to move. (**E**) Latrunculin-treated cells consistently fail to move in response to chemoattractant-stimulation. (**F**) Representative single cell volume traces show that Latrunculin-treated cells (black) lack short-term volume fluctuations but persistently maintain an elevated volume following chemoattractant stimulation. Control cells (blue) exhibit short-term volume fluctuations. (**G**) The lack of short-term volume fluctuations following latrunculin treatment is borne out across the population, with the coefficient of variation in the volume for single cells (post-swelling) being dramatically lower in Latrunculin-treated cells, suggesting that these short term volume fluctuations depend on actin-based motility.

**Author response image 2. sa3fig2:** Additional validation of swelling screen hits. (**A**) Mixed WT and CRISPR KO dHL-60 populations post-stimulation show that CA2 (black) and PI3Ky (green) KO both fail to decrease their densities as much as the WT (cyan) population following chemoattractant stimulation. Cells with negative control guides (light gray) have normal volume responses. All tubes were fractionated and aligned on the fraction containing the median of the WT population. Negative values indicate a fraction with a higher density than WT. (**B**) To validate the perturbations to cell swelling observed with FxM, primary human neutrophils were stimulated in suspension, and their volumes were measured using a Coulter counter. 20 nM fMLP was added at the 0 minute mark. Shaded regions represent the 95% confidence intervals. (**C**) PI3Kγ inhibition blocks the chemoattractant-induced volume change in primary human neutrophils, as assayed by FxM. (**D**) PI3Kγ inhibition also blocked the chemoattractant-drive shape change in human primary neutrophils, as measured by the change in footprint area in FxM (**E**) The coefficient of variation in volume for control (cyan) and iNHE1 (gold) inhibited human primary neutrophils undergoing chemokinesis are comparable, suggesting that the volume fluctuations are unchanged in moving cells upon NHE1 and PI3Kγ inhibition despite the different baseline volumes.

**Author response image 3. sa3fig3:** Additional validation of motility phenotypes. (**A-D**) Single cell tracks of primary human neutrophils responding to acute chemoattractant stimulation. Both panels show tracks of cells 15 minutes prior (left) versus 15 minutes post (right) uncaging the chemoattractant. The scale bar is 50 microns. Color saturation indicates time with tracks progressing from gray to full color. (**A**) Control cells show a large increase in movement upon uncaging, (**B**) NHE1 inhibited cells also initiate movement but to a lesser degree, (**C**) hypo-osmotic shock rescues the NHE1 motility defect. (**D**) PI3Kγ leads to a large fraction of cells failing to initiate movement. (**E**) PI3Kγ inhibition showed near complete blockage of the chemoattractant-induced motility increase in primary human neutrophils. (**F**) Control neutrophils (blue) show an increased angular alignment upon stimulation as their motility becomes directional. NHE1-inhibition (gold, iNHE1) has very little effect on this process, while PI3Kγ inhibition (green) leads to a reduction in this alignment at the population level. (**G**) For the PI3Kγ inhibited cells that start migrating, the migration-induced volume fluctuations are comparable to iNHE1 and control cells. The top panel shows the track of a representative migrating PI3Kγ inhibited cell and the bottom panel, its corresponding volume normalized to the pre-stimulation volume. The scale bar is 50 microns.

**Reviewer #2 (Public Review):**
Nagy et al investigated the role of volume increase and swelling in neutrophils in response to the chemoattractant. Authors show that following chemoattractant response cells lose their volume slightly owing to the cell spreading phase and then have a relatively rapid increase in the cell volume that is concomitant with cell migration. The authors performed an impressive genome-wide CRISPR screen and buoyant density assay to identify the regulators of neutrophil swelling. This assay showed that stimulating cells with chemoattractant fMLP led to an increase in the cell volume that was abrogated with the FPR1 receptor knockout. The screen revealed a cascade that could potentially be involved in cell swelling including NHE1 (sodium-proton antiporter) and PI3K. NHE1 and PI3K are required for chemoattractant-induced swelling in human primary neutrophils. Authors also suggest slightly different functions of NHE1 and PI3K activity where PI3K is also required to maintain chemoattractant-induced cell shape changes. The authors convincingly show that chemoattractant-induced cell swelling is linked to cell migration and NHE1 is required for swelling at the later stages of swelling since the cells at the early point work on low-volume and low-velocity regime. Interestingly, the authors also show that lack of swelling in NHE1-inhibited cells could be rescued by mild hypo-osmotic swelling strengthening the argument that water influx followed chemoattractant stimulation is important for potentiation for migration.The conclusions of this paper are mostly well supported by data and are pretty convincing, but some aspects of image acquisition and data analysis need to be clarified and extended.

We thank the reviewer for their positive appraisal of our work and pursued their suggestions for improving our paper in this revision.

Weaknesses(1) It would really help if the authors could add the missing graph for the footprint area when cells are treated with Latranculin. Graph S1F for volume changes with Lat treatment should be compared with DMSO-treated controls.

We agree that the Latrunculin condition merits more thorough investigation. To this end, we compared the volume response of human primary neutrophils to chemoattractant addition for Latrunculin B treated cells versus DMSO controls in suspension and show that there is no difference in swelling (Figure 1 – supplemental figure 2A) . This is additionally confirmed with FxM measurements with a slight undershooting of the final volume likely due to minor uptake of the excluded dye by Latrunculin treated cells (Figure 1 – supplemental figure 2B) . We have also included the requested footprint area changes in the Latrunculin treated cells as compared to controls (Figure 1 – supplemental figure 2C) . The treated cell footprints increase much less than the controls, and this is likely due to a lack of active cell spreading in the Latrunculin treated cells. The increase in footprint area observed following latrunculin treatment is within the range of what would be expected for the 2D projection of an isotropically expanding sphere fitted to the Latrunculin volume data (salmon line).

**Author response image 4. sa3fig4:** Chemoattractant-induced swelling, but not motility, is independent of actin polymerization. (**A**) Human primary eutrophils were incubated with DMSO or Latrunculin B, activated with 20 nM fMLP, and then volume responses were measured using electronic sizing via a Coulter counter. Latrunculin treatment did not alter cell swelling, indicating that actin polymerization is dispensable for the chemoattractant-induced volume increase. (**B**) Similar results were obtained using the FxM assay, showing that Latrunculin-treated cells are capable of swelling after stimulation. (**C**) The Latrunculin-treated cells also increase their footprints, albeit less so than control cells, but this is within the range of what would be expected for this degree of chemoattractant-induced volume increase (modeled by a sphere expanding an equivalent volume).

(2) The authors show inhibition of NHE1 blocked cell swelling using Coulter counter, a similar experiment should be done with PI3K inhibitions especially since they see PI3K inhibition impact chemoattractant-induced cell shape change.

Good idea. PI3Ky inhibition led to a substantial reduction in the chemoattractant-driven swelling in suspension showing the critical role of PI3K in the swelling of human primary neutrophils (Figure 3 – supplemental figure 1B) .

**Author response image 5. sa3fig5:** Additional validation of swelling screen hits. (**B**) To validate the perturbations to cell swelling observed with FxM, primary human neutrophils were stimulated in suspension, and their volumes were measured using a Coulter counter. 20 nM fMLP was added at the 0 minute mark. Shaded regions represent the 95% confidence intervals.

(3) It would be more convincing visually if the authors could also include the movie of cell spreading (footprint) and then mobility with PI3K inhibition.

Included as suggested. We agree this is a more compelling way to present the data (Figure 4 – supplemental figure 1A-D,G)

**Author response image 6. sa3fig6:** Additional validation of motility phenotypes. (**A-D**) Single cell tracks of primary human neutrophils responding to acute chemoattractant stimulation. Both panels show tracks of cells 15 minutes prior (left) versus 15 minutes post (right) uncaging the chemoattractant. The scale bar is 50 microns. Color saturation indicates time with tracks progressing from gray to full color. (**A**) Control cells show a large increase in movement upon uncaging. (**D**) PI3Kγ leads to a large fraction of cells failing to initiate movement. (**E**) PI3Kγ inhibition showed near complete blockage of the chemoattractant-induced motility increase in primary human neutrophils. (**G**) For the PI3Kγ inhibited cells that start migrating, the migration-induced volume fluctuations are comparable to iNHE1 and control cells. The top panel shows the track of a representative migrating PI3Kγ inhibited cell and the bottom panel, its corresponding volume normalized to the pre-stimulation volume. The scale bar is 50 microns.

(4) It is not clear how cell spreading and later volume increase are linked to overall mobility of neutrophils. Are authors suggesting that cell spreading is not required for cell mobility in neutrophils?

We did not mean to imply that cell spreading is not required for neutrophil motility. We take advantage of the fact that we can inhibit cell swelling without inhibiting spreading to investigate the specific role of swelling on migration ( Figure 4) . Conversely, cell spreading on a substrate is not required for chemoattractant-induced cell swelling, as chemoattractant-induced swelling occurs in latrunculin-treated cells (Figure 1 – supplemental figure 2A-C) . However, these latrunculin-treated cells are not able to migrate, at least not in the context studied here (Figure 1 – supplemental figure 2 D-E) . Cell spreading and swelling are likely both critical contributors to neutrophil motility, but their relative importance is dependent on the migratory context. The single cell volume fluctuation analysis indicates that migration-associated spreading and shape changes have large impacts on cell volume ( Figure 1 F) . These fluctuations are asynchronous, obscuring their observation at the population level, but the single cell traces clearly demonstrate them and their correlation with movement.

( 5) Volume fluctuations associated with motility were impacted by NHE1 inhibition at the baselines, what about PI3K inhibitions? Does that impact the actual fluctuations?

PI3K inhibition causes a significant fraction of cells to stop migrating (Figure 4 – supplemental figure 1D) , but among those that do move, they are still able to fluctuate in volume (Figure 4 – supplemental figure 1G) .

**Author response image 7. sa3fig7:** Additional validation of motility phenotypes. (**G**) For the PI3Kγ inhibited cells that start migrating, the migration-induced volume fluctuations are comparable to iNHE1 and control cells. The top panel shows the track of a representative migrating PI3Kγ inhibited cell and the bottom panel, its corresponding volume normalized to the pre-stimulation volume. The scale bar is 50 microns.

In contrast, latrunculin abolishes the volume fluctuations that normally accompany migration (Figure 1 – supplemental figure 2F-G) . These data suggest that movement/spreading itself is the driver of the rapid volume fluctuations. In contrast, the sustained volume increase following chemoattractant stimulation is independent of shape change and still occurs in latrunculin-treated cells.

**Author response image 8. sa3fig8:** Chemoattractant-induced swelling, but not motility, is independent of actin polymerization. (**F**) Representative single cell volume traces show that Latrunculin-treated cells (black) lack short-term volume fluctuations but persistently maintain an elevated volume following chemoattractant stimulation. Control cells (blue) exhibit short-term volume fluctuations. (**G**) The lack of short-term volume fluctuations following latrunculin treatment is borne out across the population, with the coefficient of variation in the volume for single cells (post-swelling) being dramatically lower in Latrunculin-treated cells, suggesting that these short term volume fluctuations depend on actin-based motility.

(6) It would really help if the authors compared similar analyses and drew conclusions from that, for example, it is unclear what the authors mean by they found no change in the angular persistence of WT and NHE1 inhibited cells which is in contrast to PI3K inhibition since they do not really have an analysis for angular persistence in PI3K inhibited cells. (S4A and S4B).

Thanks for catching this oversight in these experiments that we previously performed but neglected to include in the initial submission. We now include plots for angular persistence, velocity, and footprint size for the PI3K-gamma-inhibited cells. The results show that PI3K-gamma inhibition interferes both with swelling (Figure 3 – supplemental figure 1B-D) and motility (Figure 4 – supplemental figure 1D-F) , which aligns with its role upstream of the other hits identified in our screen.

**Author response image 9. sa3fig9:** Additional validation of motility phenotypes. (**A-D**) Single cell tracks of primary human neutrophils responding to acute chemoattractant stimulation. Both panels show tracks of cells 15 minutes prior (left) versus 15 minutes post (right) uncaging the chemoattractant. The scale bar is 50 microns. Color saturation indicates time with tracks progressing from gray to full color. (**A**) Control cells show a large increase in movement upon uncaging, (**B**) NHE1 inhibited cells also initiate movement but to a lesser degree, (**C**) hypo-osmotic shock rescues the NHE1 motility defect. (**D**) PI3Kγ leads to a large fraction of cells failing to initiate movement. (**E**) PI3Kγ inhibition showed near complete blockage of the chemoattractant-induced motility increase in primary human neutrophils. (**F**) Control neutrophils (blue) show an increased angular alignment upon stimulation as their motility becomes directional. NHE1-inhibition (gold, iNHE1) has very little effect on this process, while PI3Kγ inhibition (green) leads to a reduction in this alignment at the population level. (**G**) For the PI3Kγ inhibited cells that start migrating, the migration-induced volume fluctuations are comparable to iNHE1 and control cells. The top panel shows the track of a representative migrating PI3Kγ inhibited cell and the bottom panel, its corresponding volume normalized to the pre-stimulation volume. The scale bar is 50 microns.